# Effect of Deacidification Treatment on the Flavor Quality of Zaosu Pear–Kiwifruit Wine

**DOI:** 10.3390/foods11142007

**Published:** 2022-07-07

**Authors:** Xueshan Yang, Xinyuan Song, Liu Yang, Jie Zhao, Xia Zhu

**Affiliations:** 1College of Food Science and Engineering, Gansu Agricultural University, Lanzhou 730070, China; yangxs@gsau.edu.cn (X.Y.); s834263771@163.com (X.S.); 18781280131@163.com (L.Y.); z17361596120@163.com (J.Z.); 2Gansu Key Lab of Viticulture and Enology, Lanzhou 730070, China

**Keywords:** pear–kiwifruit wine, *Oenococcus oeni*, malolactic fermentation, chemical composition, sensory evaluation

## Abstract

Conventional pear–kiwifruit wine has a bland flavor and sour taste, because of excessive l-malic acid content and, consequently, little consumer appeal. An *Oenococcus oeni* strain, GF-2, has good malolactic fermentation (MLF) performance and high glucosidase activity. Through a Box–Behnken design, the optimum MLF parameters for deacidification by GF-2 were determined: initial pH of 3.4, 5% *v/v* inoculation, and temperature of 20 °C, which reduced the malic acid content by 98.3%. The changes in the content of organic acids, polyphenols, and aromatic compounds after MLF were compared with chemical deacidification. MLF significantly decreased the total concentration of organic acids by 29.7% and promoted the accumulation of aromatic esters, higher alcohols, and terpenoids, but chemical deacidification markedly decreased aromatic compound content by 59.8%. MLF wine achieved the highest sensory scores for aroma, taste, and overall acceptability. Therefore, MLF with *O. oeni* GF-2 has great potential to markedly improve the quality of commercial pear–kiwifruit wine.

## 1. Introduction

Pear (*Pyrus communis* L.) is an economically important fruit crop, which is widely distributed worldwide because of its adaptability to different climatic conditions. In 2019, China’s pear output reached 17 million tons, of which the Zaosu pear (*Pyrus bretschneideri Rehder*) output was close to 1 million tons [1]. However, the fresh fruit market has reached its saturation point, significantly reducing the price of Zaosu pears; as a consequence, orchardists are increasingly processing pears into value-added products, among which perry, or pear wine, appears to have promising market prospects. However, the commercial development of Zaosu pear wine is hindered by its bland flavor and aroma, which are not appealing to consumers [2]. The green kiwifruit (*Actinidia chinensis*) is extensively planted in China, with an output of nearly 2.2 million tons in 2019 [3]. The addition of kiwifruit juice to pear wine fermentation is a possible solution to improve the flavor and consumer appeal, because of its richness in flavor precursors such as amino acids, sugars, and proteins [4,5].

In a previous study, however, we found that wine prepared from a mixture of pear and kiwifruit juice was unacceptably sour and astringent, because of its high content of malic, quinic, and tartaric acids, derived from the kiwifruit juice [2]. The usual method to alleviate excessive acidity is deacidification by chemical or biological processing. Unfortunately, chemical acid reduction is not effective in all cases and may reduce the stability of the wine. A biological deacidification method, however, such as malolactic fermentation (MLF), can significantly reduce the concentration of malic acid and total acidity in wines, without negatively impacting the flavor quality [6,7,8,9]. Although the application of MLF in grape wine production has been thoroughly studied, its effect on the flavor profile and acidity of pear–kiwifruit wine is largely unknown. 

In addition to acid reduction, MLF can improve the aroma and taste of fermented beverages, such as Negroamaro wine [6], durian wine [10], Pinot noir wine [11], and Shiraz wine [12]. Recently, we employed an autochthonous *Oenococcus oeni* (*O. oeni*) strain GF-2, with high β-d-glucosidase and diglycosidase activity, to obtain an aromatic Chardonnay wine via MLF, after a normal alcoholic fermentation [13]; however, there has been little research on modification of fruit wine flavor by MLF with this strain.

The aim of this study was to compare the effects of chemical deacidification and MLF by *O. oeni* strain GF-2 on the flavor enhancement and organoleptic properties of Zaosu pear–kiwifruit wine. Through a combination of chemical compositional and sensory analysis, the regulatory mechanism of the formation and evolution of organic acids, polyphenols, and aromatic compounds on the flavor of pear–kiwifruit wine was identified. These findings established a foundation for the production of pear–kiwifruit wine with strong consumer appeal.

## 2. Materials and Methods

### 2.1. Materials and Chemical Reagents 

Zaosu pears and Xuxiang kiwifruits were from a local supermarket (Lanzhou, Gansu, China) and were selected for optimal ripeness and the absence of damage or spoilage. The analytical standards used in the organic acid and polyphenol analysis, oxalic, tartaric, l-malic, lactic, citric, succinic, quinic, protocatechuic, chlorogenic, and gallic acids, as well as epicatechin, catechin, caffeic acid, and phloretin, were from Shanghai Yuanye Biotechnology Co., Ltd. (Shanghai, China). The aromatic compound standards, ethyl acetate, ethyl caproate, ethyl caprylate, isoamyl acetate, hexyl acetate, *n*-pentanol, phenylethanol, *n*-hexanol, *n*-heptanol, β-citronellol, geraniol, linalool, nerol, β-damascenone, and 2-octanol (internal standard), were from Sigma-Aldrich (St. Louis, MO, USA).

### 2.2. Preparation of Pear Juice and Kiwifruit Juice

The selected fresh fruits were washed with tap water, and then drained, peeled, cut into small pieces, and juiced with a laboratory blender (DS-1, Shanghai Precision Instruments Co., Ltd., Shanghai, China). The SO_2_ concentration in the juice was adjusted to 60 mg/L by adding KHSO_3_. Pear juice was treated with 100 mg/L pectinase (100,000 U/g, Shanghai Yuanye Biotechnology Co., Ltd., Shanghai, China) at 40 °C for 3 h, while kiwifruit juice was hydrolyzed by 80 mg/L pectinase at 50 °C for 5 h, followed by centrifugation at 4000× *g* for 15 min at 4 °C [14,15]. The resulting pear juice (9.10° Brix, 5.05 g/L, pH 4.06) and kiwifruit juice (11.47° Brix, 14.62 g/L, pH 3.04) were blended at a 60:40 ratio, as described previously [2]. 

### 2.3. Fermentation of Pear–Kiwifruit Wine

#### 2.3.1. Microorganisms and Starter Culture Preparation 

*Saccharomyces cerevisiae* ES488 (Novara, Novara, Italy) and *Metschnikowia pulcherrima* 346 (Lallemand Inc., Quebec, QC, Canada) were used for the alcoholic fermentation, whereas an *O. oeni* autochthonous GF-2 strain isolated from spontaneously fermented grape juice in the Hexi Corridor region of Gansu Province [16] was used for the MLF. The selected freeze-dried yeast cultures were rehydrated using 10 times their volume of distilled water at 37 °C for 15 min (*S. cerevisiae* ES488) or 28 °C for 20 min (*M. pulcherrima* 346), and then activated with the same volume of mixed juice at 25 °C for 15 min [17]. The prescreened *O. oeni* GF-2 was stored in glycerol (40%, *v*/*v*) at −80 °C. Prior to inoculation, this strain was rejuvenated on sterilized (121 °C, 20 min) acid tomato juice agar medium (ATB) at 28 °C for 7 days, followed by anaerobic culture in ATB liquid medium (10 g/L glucose, 10 g/L peptone, 5 g/L yeast extract, 0.2 g/L MgSO_4_·7H_2_O, 0.05 g/L MnSO_4_·4H_2_O, 0.5 g/L cysteine hydrochloride, 25% *v/v* tomato juice, pH 4.8) to the logarithmic growth phase (OD600 of approximately 1.20). After proliferation, the cells were collected by centrifugation at 4000× *g* for 10 min, washed twice with sterile distilled water, and resuspended at a density of 1 × 10^7^ CFU/mL.

#### 2.3.2. Processing of Pear–Kiwifruit Wine

The winemaking was carried out on a laboratory scale as described previously [17]. The activated *M. pulcherrima* 346 was inoculated into pear–kiwifruit (60:40, 8.0 L) juice at a dosage of 0.3 g/L, and then *S. cerevisiae* ES488 (0.2 g/L) was inoculated 45 h later to continue fermentation at 22 °C until the concentration of reducing sugar was <4.0 g/L and remained unchanged for 3 days. The fermentation medium was then filtered through a 0.22 µm membrane and inoculated with the preculture of *O. oeni* GF-2 (3%, *v*/*v*) for MLF at 20 °C. When the concentration of l-malic acid remained constant, MLF was considered to be finished. All the fermentations were performed in triplicate. 

#### 2.3.3. Optimization of MLF Parameters

A single-factor test was used to evaluate the effects of initial pH (3.00, 3.20, 3.40, 3.60, or 3.80), the amount of *O. oeni* inoculum (3%, 4%, 5%, 6%, or 7% *v*/*v*), and MLF temperature (16, 18, 20, 22, or 24 °C) on the degradation rate of l-malic acid in fruit wine samples. These trials indicated that these three parameters were suitable as independent variables to optimize the MLF deacidification action, using a Box–Behnken design. The total conversion of l-malic acid was calculated as follows:(1)$$Y\;(\%)=\frac{A-B}{A}\;\times \;100,$$
where Y (%) is total conversion of l-malic acid, and A and B represent the initial and residual concentrations of l-malic acid in pear–kiwifruit wine before and after MLF, respectively. 

### 2.4. Chemical Deacidification Treatment

As described previously [2], 1.0 g/L Na_2_CO_3_, 1.0 g/L K_2_CO_3_, and 5.0 g/L KHC_4_H_4_O_6_ (potassium hydrogen tartrate) were added to reduce the titratable acidity (TA) of pear–kiwifruit wine to 5.26 g/L. 

### 2.5. Determination of Chemical Composition

The pH and soluble solid content were determined directly, using a pH meter (PHS-3C, Shanghai Leici Instrument Factory, Shanghai, China) and a refractometer (PAL-2, Atago Co., Ltd., Tokyo, Japan). Measurements of reducing sugars, titratable acid (TA), volatile acids, alcohol content, and total phenols were performed following the official methods for musts and wines [18]. 

### 2.6. Determination of Individual Organic Acids and Polyphenols

Individual organic acids were analyzed by HPLC as described previously [19], with minor modifications. Each fruit wine sample was centrifuged at 10,000× *g* for 10 min, and then the supernatant was filtered through a 0.45 µm PVDF syringe filter. For quantification of organic acids, a 20 µL sample was injected into an LC-ZOA HPLC system (Shimadzu, Kyoto, Japan), fitted with an Agilent TC-C18 column (4.6 mm × 250 mm, 5 µm; Agilent Technologies, Santa Clara, CA, USA) maintained at 30 °C and an ultraviolet (UV) detector set at 210 nm. The isocratic mobile phase was 0.06 mol/L potassium dihydrogen phosphate, pH 2.55, at a flow rate of 0.5 mL/min. 

The same HPLC system, fitted with an Agilent Zorbax Eclipse SB-C18 column (4.6 mm × 250 mm, 5 µm) and detected at 280 nm was used to determine polyphenols. The column temperature was maintained at 25 °C, and the injection volume was 10 µL. The mobile phases were 1% *v/v* acetic acid in deionized water (A) and 1% *v/v* acetic acid in methanol (B), at a flow rate of 1 mL/min. Gradient elution was performed as follows: 0 min, 10% B; 10 min, 26% B; 25 min, 40% B; 45 min, 65% B; 55 min, 95% B; 58 min, 10% B; 61 min, 10% B. Identification and quantification were performed with reference to authentic standards and their calibration curves (Appendix A). 

### 2.7. Analysis of Volatile Composition

Volatiles in pear–kiwifruit wines were analyzed by headspace solid-phase microextraction (HS-SPME), combined with gas chromatography–mass spectrometry (GC–MS) as described previously [17]. Wine samples (8 mL) were mixed with NaCl (2.4 g) and an internal standard solution (2-octanol in methanol, 81.06 mg/L, 10 μL) in a headspace vial, and then incubated at 40 °C for 15 min, in the presence of an SPME fiber coated with 50/30 μm DVB/CAR/PDMS (57329-U; Supelco, Bellefonte, PA, USA). The volatiles were thermally desorbed from the fiber at 250 °C in splitless mode for 5 min, in a TRACE 1310 GC (Thermo Fisher Scientific, Waltham, MA, USA), fitted with a DB-WAX column (60 m × 0.25 mm, 0.25 µm; J & W Scientific, Santa Clara, CA, USA) with helium as the carrier gas at 1 mL/min. The temperature program was set at 40 °C for 5 min, increased at 3.5 °C/min to 180 °C, held for 15 min, increased at 6 °C/min to 220 °C, and held for 10 min. Volatiles were detected and identified by mass spectrometry in electron impact (EI) mode at 70 eV with an *m*/*z* scan range of 35–350. The ion source and connecting rod temperatures were 250 °C and 180 °C, respectively. The retention times of standards and matching of mass spectra to the NIST 17 library were employed to identify volatile compounds. Specific components were quantified by using a calibration curve or by comparing peak areas to that of the internal standard. 

### 2.8. Sensory Evaluation

The check-all-that-apply (CATA) method was used to assess the characteristics of pear–kiwifruit wine samples. After explaining the tasting procedure to all participants, those willing to continue the sensory assessment signed the consent form. A total of 36 untrained volunteers (24 males and 12 females, 23–56 years old) were recruited for evaluation of appearance (color and clarity), flavor (aroma, taste, and mouthfeel), and overall acceptability, using a nine-point hedonic scale, where 1 indicated strongly dislike and 9 indicated strongly like. The panelists were also asked to select appropriate attributes from a questionnaire containing 25 sensory terms to describe the product as previously described [20], with some modifications. During evaluation, coded samples (30 mL) were served in ISO wine glasses, in a balanced, randomized order under white light, with the temperature of the separate sensory booths controlled at 22–24 °C.

### 2.9. Statistical Analysis

Significant differences in chemical composition and sensory characteristics between samples were determined by one-way analysis of variance (ANOVA) and Tukey’s (HSD) test, using SPSS 19.0 software (SPSS Inc., Chicago, IL, USA). Principal component analysis (PCA) was performed using OriginPro 2018 (OriginLab, Inc., Northampton, MA, USA) to distinguish the compositions of pear–kiwifruit wines produced by different deacidification processes. The response surface optimization results were analyzed with Design-Expert 8.0.6 software (Stat-Ease, Minneapolis, MN, USA). 

## 3. Results and Discussion

### 3.1. Optimization of MLF Parameters for Pear–Kiwifruit Wine

#### 3.1.1. Effects of MLF Parameters on Degradation Rate of l-Malic Acid in Wine Samples

To optimize MLF deacidification by *O. oeni* GF-2, three parameters were selected, i.e., initial pH, inoculation amount (% *v*/*v*), and fermentation temperature. The total conversion of l-malic acid in pear–kiwifruit wine increased markedly from pH 3.0 to 3.4, and then increased more slowly, reaching a peak (94.2%) at pH 3.8 (Figure 1A). It is generally considered that a low pH would inhibit the growth of *O. oeni,* by stabilizing SO_2_ and maximizing its antibacterial activity [21,22,23]. The conversion efficiency of l-malic acid showed a clear convex trend over the tested temperature range (Figure 1B), reaching a peak (92.3%) at 20 °C. The degradation of l-malic acid was maximal at a 7% (*v*/*v*) inoculation of *O. oeni* GF-2, but was not significantly lower at a 5% inoculation (Figure 1C). Overall, it appeared that *O. oeni* GF-2 could adapt well to the conditions in fruit wine. 

#### 3.1.2. Predictive Model Fitting of Deacidification Efficiency

According to the results of single-factor tests and estimated production costs, combined with the risk of microbial spoilage, the levels of the three independent variables, initial pH (A), inoculation amount (B), and MLF temperature (C), were subjected to a Box–Behnken design, to maximize the degradation of l-malic acid as the dependent response (Y). A total of 17 experiments were performed (Table 1), and a regression model was obtained to express the relationship between the investigated variables (A, B, and C) and the response value (Y) as follows:Y = 97.79 − 1.10A − 0.36B + 0.41C + 0.15AB − 0.31AC − 0.61BC − 4.73A^2^ − 3.91B^2^ − 1.87C^2^.(2)

The *F*-value obtained from ANOVA was 103.36, with a *p*-value <0.0001, and this nonsignificant lack of fit implied that the model was very closely related to MLF efficiency, and that the regression analysis was effective (Table 2). Furthermore, the coefficient of determination (*R*^2^), the adjusted *R*^2^, and the coefficient of variation (*CV*) were 0.9925, 0.9829, and 0.51%, respectively, indicating that the model could accurately predict the percentage degradation of l-malic acid in pear–kiwifruit wine during MLF with *O. oeni* GF-2. The one-order term coefficient of initial pH (A) and all quadratic term coefficients were very significantly related to deacidification efficiency (*p* < 0.01), whereas the inoculation amount (C) and its interaction with temperature had a significant effect (*p* < 0.05) on the percentage degradation of l-malic acid in wine samples (Figure 2C). The other terms were not significantly related to MLF efficiency (Figure 2A,B). 

The optimal combination of MLF parameters predicted by the regression equation was as follows: initial pH of 3.4, inoculation amount of 5%, and temperature of 20 °C; under these conditions, the degradation of l-malic acid was predicted to be 97.9%. An 8.0 L-scale MLF confirmatory test was performed in triplicate under the above conditions, achieving an average 98.3% reduction in l-malic acid content, very close to the predicted value. 

### 3.2. Chemical Composition of Pear–Kiwifruit Wine

#### 3.2.1. Physicochemical Properties of Three Wine Samples

The composition of major components in kiwifruit–pear wine was determined (Table 3). The final reducing sugar content in MLF wine, prepared under the optimized conditions above (2.34 g/L), was significantly lower than that in non-MLF-treated wines, because of the sugar consumption by *O. oeni* GF-2. The alcohol content ranged from 6.41% to 6.57% (*v*/*v*), and there was no significant difference between non-MLF-treated wines. After MLF and chemical deacidification, the TA of the alcoholic fermented wines decreased by 15.1% and 28.7%, respectively, and consequent increases in pH were observed (Table 3). It is generally considered that a volatile acid concentration of more than 0.8 g/L imparts an undesirable pungent smell to wines [24,25]. Although the use of MLF with *O. oeni* GF-2 increased the volatile acid concentration to 0.21 g/L, this value was well below the above threshold [25]. 

#### 3.2.2. Organic Acid Content of Pear–Kiwifruit Wine after Deacidification

We previously identified seven organic acids in pear–kiwifruit juice and wines made from it (Figure 3A). l-Malic acid (1.97 g/L) was the most abundant organic acid in the mixed juice, followed by quinic acid (1.53 g/L) and tartaric acid (1.43 g/L), depending on the specific fruit varieties used (Appendix A). After alcoholic fermentation, the concentrations of citric, succinic, and lactic acids increased significantly (*p* < 0.05), whereas those of oxalic, l-malic, quinic, and tartaric acids decreased (Figure 3A). 

As expected, MLF with *O. oeni* GF-2 greatly reduced l-malic acid (by 97.6%) and correspondingly increased the concentration of lactic acid from 0.64 to 1.23 g/L (Figure 3A), i.e., this strain had excellent fermentation performance in pear–kiwifruit wine. It appeared that the malolactic enzyme (MLE) produced by *O. oeni* GF-2 had a strong affinity for l-malic acid. The total organic acid content in MLF wine was 29.7% lower than that in untreated wine, mainly resulting from significant reductions in succinic and tartaric acid, in addition to that of l-malic acid (Appendix A). Unlike succinic acid, tartaric acid is not usually converted by *O. oeni*, but precipitates as potassium hydrogen tartrate [26]. MLF also reduced the concentration of astringent quinic acid by up to 0.2 g/L, indicating that *O. oeni* GF-2 could balance the wine taste by alleviating excessive astringency, as well as sourness. 

The chemical deacidification with 1.0 g/L Na_2_CO_3_, 1.0 g/L K_2_CO_3_, and 5.0 g/L KHC_4_H_4_O_6_ significantly reduced the concentrations of all detected organic acids (*p* < 0.05), except l-malic acid, and lactic acid completely disappeared (Figure 3A). Chemically deacidified wine had a lower total concentration (4.35 g/L) of organic acids than MLF wine, because of a marked decrease in lactic, oxalic, and tartaric acids. 

#### 3.2.3. Polyphenol Content of Pear–Kiwifruit Wine after Deacidification 

Seven polyphenols were identified in pear–kiwifruit juice (Figure 3B), including catechin (10.17 µg/mL), chlorogenic acid (8.26 µg/mL), and epicatechin (6.33 µg/mL). After alcoholic fermentation, the concentrations of most polyphenols, including the three above, increased by at least 20%, which can be attributed to the extraction of polyphenols from the fruit pomace during fermentation. β-Glucosidase produced by lactic acid bacteria, such as *O. oeni,* can increase the content of chlorogenic acid by hydrolyzing chlorogenic acid glycosides [27]; *O. oeni* GF-2 has a strong capacity to produce β-glucosidase [13]. Consequently, the highest concentration of chlorogenic acid (11.06 µg/mL) was observed in MLF wine (Figure 3B), whereas the concentrations of epicatechin, catechin, and phloretin decreased by 15.9%, 12.5%, and 26.2%, respectively, after MLF (Appendix A), in agreement with reports that MLF significantly reduced the content of catechin in red wine, resulting from metabolism by *O. oeni* [28,29]. However, the content of epicatechin decreased, in contrast to a previous report [29]; a possible reason for decreased epicatechin is that it may be involved in polymerization during MLF.

The concentrations of gallic and caffeic acids decreased during alcoholic fermentation, but increased during MLF (Figure 3B), which may have resulted from a different balance among oxidation, biotransformation induced by *O. oeni* GF-2, and hydrolysis of higher oligomers. There was no significant difference in protocatechuic acid concentration after MLF, whereas it was undetectable after chemical deacidification. The content of other individual polyphenols decreased by at least 18.6% after chemical deacidification, except for gallic acid, resulting in a lower total polyphenol concentration than that in MLF wine (Figure 3B). Chemical deacidification resulted in low total organic acids, which may decrease the stability and, consequently, the total content of polyphenols. Overall, both chemical and biological deacidification modified the contents of organic acids and polyphenols in pear–kiwifruit wine. 

### 3.3. Effect of Deacidification on Aromatic Profile of Pear–Kiwifruit Wine 

A total of 77 volatiles were detected in all wine samples, from the following chemical classes: esters, higher alcohols, acids, terpenoids, norisoprenoids, aldehydes, ketones, and volatile phenols (Appendix A). Pear–kiwifruit juice and alcoholic fermented wine contained 59 and 61 volatile aromatic compounds, respectively, whereas MLF wine contained 64, and chemically deacidified wine contained 35 (Figure 4A). The total content of aromatic compounds in alcoholic fermented wine was higher than in chemically deacidified wine, but significantly lower than in MLF wine (Figure 4B). 

Esters were the dominant class of aromatic compounds in mixed juice and all wines, especially ethyl esters and acetate esters (Appendix A); isoamyl acetate, ethyl caproate, ethyl octanoate, and ethyl decanoate were the most abundant aromatic compounds in all treatments, accounting for >70% of the total ester concentration (Table 4); the highest total ester concentration (11,598 µg/L) was in MLF wine (Figure 4C). In agreement with these findings, MLF with *O. oeni* also enhanced the total ester content of red wine [11,25,30,31]. In this study, the contents of carboxylic acid and alcohol esterification substrates did not decrease after the formation of esters; it appears that the high glycosidase activity produced by GF-2 [13] promoted ester accumulation via the hydrolysis of glycoconjugates. In contrast, a significant decrease in ester content after MLF with a different *O. oeni* strain was observed [30,32], indicating that the evolution of esters during MLF with *O. oeni* is strain-dependent and depends on the balance among esterification, ester hydrolysis, and glycoconjugate hydrolysis. Notably, ethyl lactate, a characteristic ester produced during MLF, was present only in MLF wine, and the total concentration of pear aromatic esters (i.e., methyl (2*E*,4*Z*)-deca-2,4-dienoate, ethyl (2*E*,4*Z*)-deca-2,4-dienoate, heptyl acetate, ethyl caprylate, and hexyl acetate) increased by 8.3% after MLF (Table 4). After chemical deacidification, 20 esters disappeared (Appendix A), the contents of others significantly decreased, and the total ester content decreased by about two-thirds (Figure 4C). This result confirms that the precipitation of organic acid salt formed in the process of chemical deacidification caused the loss of aromatic compounds in fruit wine [33].

Alcohols (other than ethanol) were the second largest group of volatiles in mixed juice and pear–kiwifruit wines, of which 1-pentanol, 1-hexanol, and 2-phenylethanol were dominant and contributed >94% of the total alcohol content (Appendix A). The total alcohol concentration increased by 13.3% after MLF (to 6125 µg/L) and decreased by ~60% after chemical deacidification (to 2181 µg/L) (Figure 4D). During alcoholic fermentation, phenylalanine metabolism produces 2-phenylethanol, which has a rose/honey aroma [25,34,35]; this, however, cannot explain the accumulation of this compound in this study, considering the absence of sugar and nitrogen sources occurring in MLF. It is possible that glycoconjugate precursors in pear–kiwifruit wines were hydrolyzed by *O. oeni* GF-2 glycosidase, to release 2-phenylethanol. There was a significant increase previously found in 2,3-butanediol during MLF with *O. oeni* [6,12], in agreement with the finding here that the concentration of 2,3-butanediol increased eightfold after MLF (Table 4), conferring a complexity of aroma to the wine. The decreased concentration of citric acid after MLF (Figure 3A) suggested that GF-2 increased 2,3-butanediol formation through degradation of diacetyl, a product of citric acid metabolism. 

The total concentrations of volatile fatty acids in all samples were below the off-flavor threshold of 20 mg/L, and no significant difference was observed between wines before and after MLF (Figure 4E). This indifference may be attributed to the different responses of individual compounds (Appendix A) to biosynthesis and esterification during MLF. Octanoic acid was the most abundant fatty acid in all wines, with the highest concentration (341 µg/L) in MLF wine (Table 4), in agreement with previous reports [12,36]. Octanoic acid decreased by 79.2% after chemical deacidification and, consequently, the total fatty acid content decreased. We speculate that the CO_2_ formed by deacidification agent treatment accelerates the volatilization loss of this compound.

MLF significantly increased the total concentration of terpenoid alcohols (80.7 µg/L), whereas chemical deacidification decreased it (Figure 4F). Linalool, citronellol, and α-terpineol are important contributors to the fruity aroma of pear–kiwifruit wine, accounting for 60% of the terpenoids after alcoholic fermentation (Appendix A). Presumably because of glycoconjugate hydrolysis, the content of the three main terpenoid alcohols increased significantly after MLF (Table 4), which is consistent with the high β-D-glucosidase and diglycosidase activities produced by *O. oeni* GF-2 [13]. In contrast, chemical deacidification decreased the concentrations of all terpenoids (Appendix A); the pH change after deacidification may have reduced the stability of terpenoids, such that degradation reduced their accumulation.

### 3.4. Sensory Characteristics of Pear–Kiwifruit Wines

Chemical treatment or MLF deacidification had little effect on the color or clarity (Figure 5A,B, *p* > 0.05) and mean taste scores of the wines, while the scores of the samples after both treatments were significantly higher than those of the wines from the initial alcoholic fermentation. The MLF wine scored highest for aroma, taste, and overall acceptability. Chemical deacidification improved the taste and overall acceptability scores of wines, but reduced the aroma scores (Figure 5B). Overall, MLF wine had a superior sensory profile. 

Principal component analysis (PCA) was performed to further characterize the check-all-that-apply (CATA) sensory descriptors of pear–kiwifruit wine (Appendix A). The first and second principal components (PC1 and PC2) explained 49.9% and 35.6% of the variation, respectively (85.5% combined) (Figure 5C); the selected attributes provided a clear distinction among untreated wine, MLF wine, and chemically deacidified wine. Before deacidification, the wine was characterized as “fruity”, “acidic”, “astringent”, and “sharp”, whereas MLF wine, located at the positive end of PC1, was described as “pear-like”, “fruity”, “acidic/fresh” (balance between sweetness and acidity), “diverse”, and “floral” (Appendix A). This result was consistent with the large reduction in l-malic and quinic acids (Figure 3A), as well as the increase in aromatic compounds (Appendix A), after MLF with *O. oeni* GF-2. The chemically deacidified wine was characterized as “acidic/fresh”, “buttery”, and “bitter” (Appendix A); the marked decrease in acidity appears to have rendered the bitterness of the wine more noticeable.

## 4. Conclusions

This study investigated the effect of deacidification, by malolactic fermentation (MLF) or a chemical method, on the chemical composition of pear–kiwifruit wine. Both deacidification treatments markedly decreased the total organic acid concentration. The total phenol and volatile concentrations decreased after chemical deacidification, resulting in a low sensory score for aroma quality. Under the optimal parameters for MLF determined for *O. oeni* autochthonous strain GF-2, i.e., initial pH of 3.4, inoculation amount of 5%, and fermentation temperature of 20 °C, MLF increased the concentrations of aromatic esters, higher alcohols, and terpenoids, imparting a rich fruity and floral aroma to the wine. The MLF wine also achieved the highest sensory scores for aroma, taste, and overall acceptability. Therefore, MLF with *O. oeni* GF-2 has great potential to markedly improve both the bland flavor and the excessive acidity/sourness of commercial pear–kiwifruit wine.

## Figures and Tables

**Figure 1 foods-11-02007-f001:**
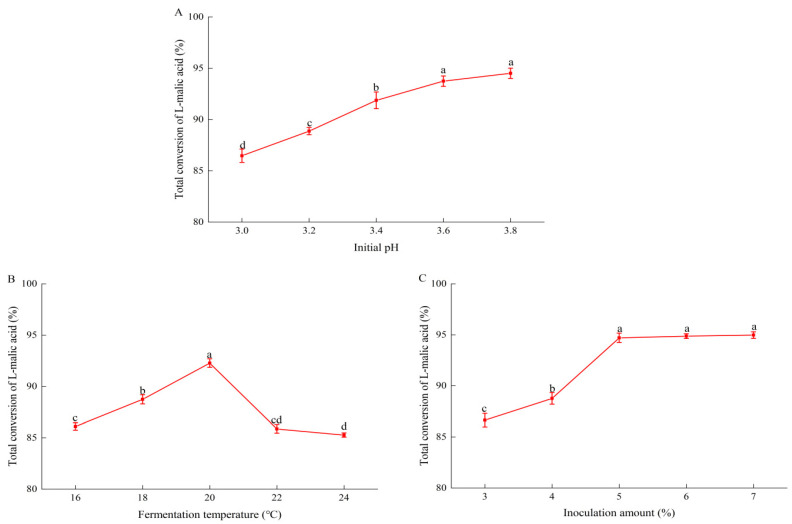
Effects of malolactic fermentation parameters by *Oenococcus oeni* strain GF-2 on total conversion of l-malic acid in pear–kiwifruit wine: (**A**) initial pH; (**B**) malolactic fermentation temperature; (**C**) amount of *O. oeni* inoculum. Error bars indicate the SD (*n* = 3). Values with different letters are significantly different (*p* < 0.05) among the samples.

**Figure 2 foods-11-02007-f002:**
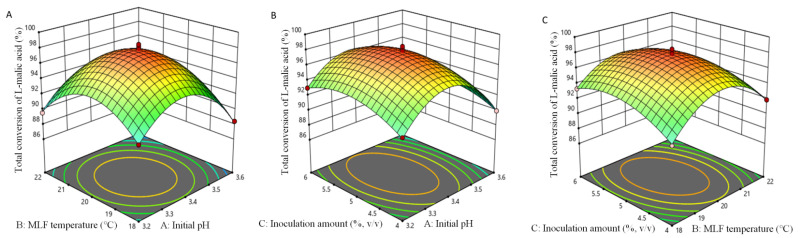
Response surface and contour plots for total conversion of l-malic acid in pear–kiwifruit wine as a function of the independent variables: (**A**) effect of the initial pH and malolactic fermentation temperature; (**B**) effect of the initial pH and inoculation amount; (**C**) effect of the malolactic fermentation temperature and inoculation amount.

**Figure 3 foods-11-02007-f003:**
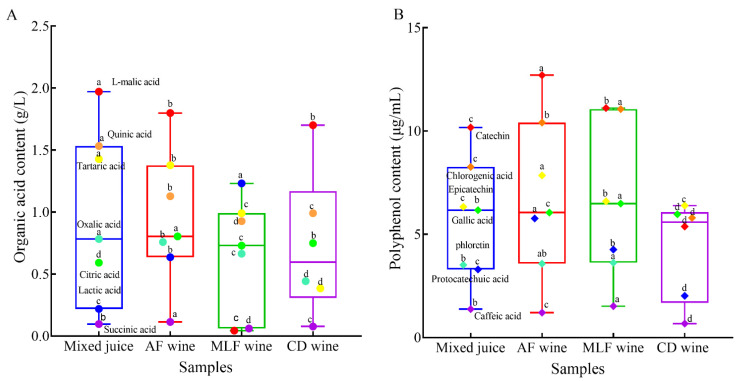
Deacidification modulates organic acid and polyphenol accumulation in pear–kiwifruit wine: (**A**) organic acid; (**B**) polyphenol. Mixed juice: pear and kiwifruit juice at the blend ratio of 60:40. AF wine: pear–kiwifruit juice co-inoculated with *Saccharomyces cerevisiae* ES488 and *Metschnikowia. pulcherrima* 346. MLF wine: AF wine inoculated with *Oenococcus oeni* strain GF-2. CD wine: wine chemically deacidified by 1.0 g/L Na_2_CO_3_, 1.0 g/L K_2_CO_3_, and 5.0 g/L KHC_4_H_4_O_6_. Data are the mean ± SD of three independent experiments. The same symbols with different letters represent significant differences (*p* < 0.05).

**Figure 4 foods-11-02007-f004:**
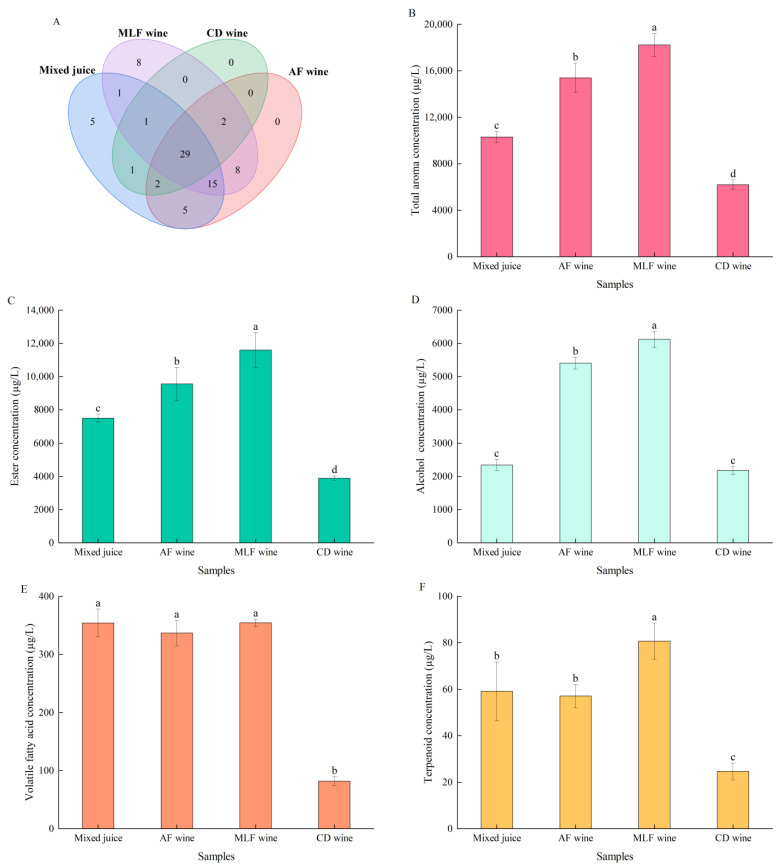
Effects of deacidification treatment on aromatic profile of pear–kiwifruit wine: (**A**) commonly and uniquely aroma compounds in pear–kiwifruit juice and corresponding wines with and without deacidification treatment; (**B**) total concentration of aroma compounds; (**C**) esters; (**D**) alcohols; (**E**) volatile fatty acids; (**F**) terpenoids. Error bars indicate the SD (*n* = 3). Values with different letters are significantly different (*p* < 0.05) among the samples.

**Figure 5 foods-11-02007-f005:**
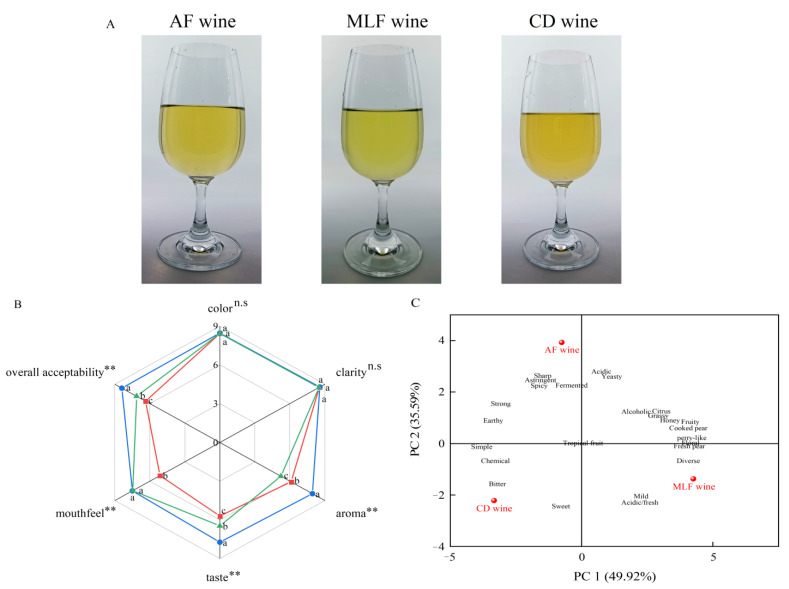
Sensory characteristics of pear–kiwifruit wine with and without deacidification treatment: (**A**) fruit wine samples; (**B**) sensory profiles obtained for AF wine (—), MLF wine (—), and CD wine (—); ns = not significant, ** *p* < 0.01, values with different letters are significantly different among the samples; (**C**) principal component analysis for check-all-that-apply attributes; *n* = 36.

**Table 1 foods-11-02007-t001:** Box–Behnken design and response values for MLF-treated pear–kiwifruit wine.

Run	Independent Variable			Response
	A Initial pH	B MLF Temperature (°C)	C ^a^ Inoculation Amount (%, *v*/*v*)	Total Conversion of l-Malic Acid (%)
1	−1 (3.20)	−1 (18)	0 (5)	90.78
2	1 (3.60)	−1	0	88.42
3	−1	1 (22)	0	89.55
4	1	1	0	87.81
5	−1	0 (20)	−1 (4)	91.70
6	1	0	−1	89.96
7	−1	0	1 (6)	93.03
8	1	0	1	90.06
9	0 (3.40)	−1	−1	91.19
10	0	1	−1	91.90
11	0	−1	1	93.34
12	0	1	1	91.60
13	0	0	0	98.16
14	0	0	0	96.93
15	0	0	0	98.46
16	0	0	0	97.85
17	0	0	0	97.54

The experiments for each MLF parameter combination were carried out in triplicate. All degradation rates of l-malic acid are expressed using the average value to eliminate the errors. ^a^ The amount of *O. oeni* GF-2 inoculum with a density of 1 × 10^7^ FU/mL.

**Table 2 foods-11-02007-t002:** Variance analysis and significance test of regression model.

Source	Sum of Squares	Degree of Freedom	Mean Square	*p*-Value *
Model	204.92	9	22.77	<0.0001
A	9.7	1	9.7	0.0003
B	1.03	1	1.03	0.0674
C	1.34	1	1.34	0.0428
AB	0.096	1	0.096	0.5301
AC	0.38	1	0.38	0.2314
BC	1.50	1	1.50	0.0349
A^2^	94.36	1	94.36	<0.0001
B^2^	64.50	1	64.50	<0.0001
C^2^	14.67	1	14.67	<0.0001
Residual	1.54	7	0.22	
Lack of fit	0.15	3	0.050	0.9282
Pure error	1.39	4	0.35	
Total	206.46	16		

*** A *p*-value < 0.05 indicates that the model terms are significant, while a *p*-value < 0.01 indicates that the model terms are extremely significant.

**Table 3 foods-11-02007-t003:** Physicochemical properties of three pear-kiwifruit wines.

Composition	AF Wine ^a^	MLF Wine ^b^	CD Wine ^c^
Total sugar (glucose, g/L)	3.93 ± 0.02 ^a^	2.34 ± 0.04 ^b^	3.87 ± 0.03 ^a^
Titration acid (tartaric acid g/L)	7.38 ± 0.03 ^a^	6.28 ± 0.05 ^b^	5.26 ± 0.02 ^c^
pH	3.36 ± 0.03 ^c^	3.50 ± 0.02 ^b^	3.72 ± 0.01 ^a^
Volatile acidity (acetic acid, g/L)	0.10 ± 0.03 ^b^	0.21 ± 0.01 ^a^	0.11 ± 0.02 ^b^
Alcohol content (%, *v*/*v*)	6.43 ± 0.05 ^b^	6.57 ± 0.05 ^a^	6.41 ± 0.05 ^b^

All values are reported as the mean (±SD) of three experiments. Values in the same row with different letters indicate a statistical difference according to Tukey’s test (*p* < 0.05). ^a^ AF wine: pear–kiwifruit juice co-inoculated with *Saccharomyces cerevisiae* ES488 and *Metschnikowia. pulcherrima* 346. ^b^ MLF wine: AF wine inoculated with *Oenococcus oeni* strain GF-2. ^c^ CD wine: wind chemically deacidified by 1.0 g/L Na_2_CO_3_, 1.0 g/L K_2_CO_3_, and 5.0 g/L KHC_4_H_4_O_6_.

**Table 4 foods-11-02007-t004:** Main aromatic compounds in pear–kiwifruit juice and corresponding wines with and without deacidification treatment.

Compounds	Aroma Compounds Concentration (μg/L)				Odor Descriptor ^a^
	Mixed Juice ^a^	AF Wine ^b^	MLF Wine ^c^	CD Wine ^d^	
Ethyl caprylate	2899 ± 116 ^a^	3129 ± 485 ^a^	3458 ± 498 ^a^	1979 ± 47 ^a^	Pineapple, pear, Fruity, floral
Isoamyl acetate	483 ± 31 ^b^	1231 ± 98 ^a^	1305 ± 166 ^a^	349 ± 18 ^c^	Fruity, banana
Ethyl hexanoate	1037 ± 98 ^a^	1014 ± 131 ^a^	1138 ± 88 ^a^	420 ± 13 ^b^	Pineapple, banana
Ethyl caprate	874 ± 79 ^c^	1485 ± 122 ^b^	2188 ± 150 ^a^	341 ± 17 ^d^	Fruity, fat, floral
Ethyl (2*E*,4*Z*)- deca-2,4-dienoate	718 ± 18 ^b^	730 ± 27 ^b^	879 ± 41 ^a^	6.3 ± 1.1 ^c^	Pear
Methyl (2*E*,4*Z*)- deca-2,4-dienoate	2.8 ± 0.3 ^b^	1.3 ± 0.1 ^b^	4.2 ± 0.3 ^a^	ND	Pear
Ethyl laurate	320 ± 18 ^b^	716 ± 21 ^a^	708 ± 5 ^a^	304 ± 6 ^b^	Fruity, fat
Hexyl acetate	503 ± 24 ^a^	425 ± 19 ^c^	477 ± 11 ^b^	210 ± 14 ^d^	Fruity, pear, floral
Ethyl butanoate	15.4 ± 4.7 ^c^	69.6 ± 4.6 ^a^	66.2 ± 8.3 ^a^	24.3 ± 0.9 ^b^	Fruity
Ethyl lactate	ND	ND	1.9 ± 0.1 ^a^	ND	Wine
Pentanol	1029 ± 142 ^b^	3264 ± 70 ^a^	3452 ± 121 ^a^	1087 ± 48 ^b^	Balsamic, fruity
Phenylethyl alcohol	528 ± 64 ^d^	1398 ± 48 ^b^	1780 ± 85 ^a^	724 ± 41 ^c^	Rose, floral
Hexanol	650 ± 42 ^a^	480 ± 35 ^a^	527 ± 17 ^a^	249 ± 16 ^b^	Grass
Isobutanol	35.6 ± 12.8 ^b^	103.4 ± 8.8 ^a^	102.7 ± 2.5 ^a^	33.0 ± 1.7 ^b^	Sweet, alcohol
1-Heptanol	50.1 ± 5.2 ^a^	51.0 ± 4.1 ^a^	54.2 ± 1.5 ^a^	42.9 ± 1.8 ^b^	Green, fruity, lemon, citrus
Octanoic acid	326 ± 20 ^a^	276 ± 17 ^b^	341 ± 5 ^a^	70.9 ± 5.3 ^c^	
2-Methylbutyric acid	16.2 ± 2.9 ^b^	45.9 ± 3.6 ^a^	ND	6.2 ± 0.4 ^c^	Cheese, sour fruity
Linalool	16.2 ± 3.5 ^b^	15.4 ± 1.5 ^b^	33.3 ± 1.8 ^a^	10.6 ± 0.6 ^c^	Rose, lavender
Citronellol	8.9 ± 2.4 ^c^	14.0 ± 0.5 ^b^	18.7 ± 0.7 ^a^	9.8 ± 0.5 ^c^	Lemon, citrus
α-Terpineol	3.8 ± 1.1 ^b^	4.3 ± 0.3 ^b^	15.1 ± 0.6 ^a^	1.9 ± 0.2 ^c^	Lilac

Overall, MLF with *O. oeni* GF-2 favored the formation of aromatic esters, higher alcohols, and terpenoids, whereas chemical deacidification resulted in a marked loss of aromatic compounds. All values are reported as the mean (±SD) of three experiments. ND: not detected. Values in the same row with different letters indicate a statistical difference according to Tukey’s test (*p* < 0.05). ^a^ Mixed juice: pear and kiwifruit juice at the blend ratio of 60:40. ^b^ AF wine: pear–kiwifruit juice co-inoculated with *Saccharomyces cerevisiae* ES488 and *Metschnikowia. pulcherrima* 346. ^c^ MLF wine: AF wine inoculated with *Oenococcus oeni* strain GF-2. ^d^ CD wine: wine chemically deacidified by 1.0 g/L Na_2_CO_3_, 1.0 g/L K_2_CO_3_, and 5.0 g/L KHC_4_H_4_O_6_.

## Data Availability

Data are contained within the article or Appendix A.

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
