# Peer review of "Effect of Deacidification Treatment on the Flavor Quality of Zaosu Pear–Kiwifruit Wine"

_foods, 2022, doi:10.3390/foods11142007_

Round 1

Reviewer 1 Report

The aim of study is insteresting and the experimental design well done and really complete, the results demonstrated the efficient of MLF to improve the sensory perception of pear-kiwifruit wine with excelent perspective for aplication at industrial level.

Minnor comment:

1. Add the color references in figure 5A.

2. English could be revised. Some parts results little confused regards the use of connectors such as "but" and "however" and could be revised. 

3. Explain better if chemical deacidification is used in vinification.

4. Figure 1 are preliminar results of figure 2, I think that figure 1 could go to supplementary material.

Reviewer 2 Report

The manuscript entitled “Effect of deacidification treatment on the flavor quality of Zaosu pear-kiwifruit wine” investigated the impact of chemical and biological deacidification of a specific fruit wine, using a selected strain of Oenococcus oeni and an experimental design with multiple parameters. Chemical analyses were carried out to assess the wine composition and the changes of volatile compounds after alcoholic and malolactic fermentations, as well as the chemical treatment. Sensory analysis determined the aroma and taste repercussions of the deacidification strategies. The article is well-written and clear, but there are some specific points to be addressed as suggested below:

1.     Lines 445-545: the whole References section needs to be revised, the numbers do not correspond to what is indicated in the text. 

2.     Lines 48-50, 84-85: in the article by Zhu et al. (2021), the strains used were ZX-1 and GF-2, but in the present study you indicate GF-1, which one did you use? Also, in Zhu et al. (2020) there is no reference to GF-1. Furthermore, Chardonnay is not an aromatic grape variety, what do you mean by aromatic wine?

3.     Lines 198-199: it is not clear how the “estimated production costs” and the “risk of microbial spoilage” were used for the determination of levels in the experimental design.

4.     Lines 243-247: is it possible to consider the same threshold of volatile acidity for wine and pear-kiwifruit wine?

5.     Lines 256-278: for the discussion of organic acids, it might be clearer to report the values in g/L.

6.     Lines 259-261 and Table S3: how do you explain the decrease of malic acid and increase of lactic acid after the AF? Was it the yeast activity or possibly some spontaneous MLF during the early stages of fermentation? Did you check for the presence of native bacteria in the juices?

7.     Lines 264-267: if you didn’t measure the gene expression, as it was not the scope of the present study, these sentences do not aggregate to the discussion.

8.     Figure 5: Please add a legend for the radar plot.
